

# Isolation and evolutionary analyses of porcine epidemic diarrhea virus in Asia

Wan Liang[1,2], Danna Zhou[1], Chao Geng[1,3], Keli Yang[1], Zhengying Duan[1], Rui Guo[1], Wei Liu[1], Fangyan Yuan[1], Zewen Liu[1], Ting Gao[1], Ling Zhao[2], Dongwan Yoo[4] and Yongxiang Tian[1]

[1] Key Laboratory of Prevention and Control Agents for Animal Bacteriosis (Ministry of Agriculture), Institute of Animal Husbandry and Veterinary Science, Hubei Academy of Agricultural Sciences, Wuhan, China
[2] Key Laboratory of Preventive Veterinary Medicine of Hubei Province, College of Veterinary Medicine, Huazhong Agricultural University, Wuhan, China
[3] College of Animal Sciences, Yangtze University, Jinzhou, China
[4] Department of Pathobiology, College of Veterinary Medicine, University of Illinois at Urbana-Champaign, Champaign, United States of America

## ABSTRACT

Porcine epidemic diarrhea virus (PEDV) is a leading cause of diarrhea in pigs worldwide. Virus isolation and genetic evolutionary analysis allow investigations into the prevalence of epidemic strains and provide data for the clinical diagnosis and vaccine development. In this study, we investigated the genetic characteristics of PEDV circulation in Asia through virus isolation and comparative genomics analysis. APEDV strain designated HB2018 was isolated from a pig in a farm experiencing a diarrhea outbreak. The complete genome sequence of HB2018 was 28,138 bp in length. Phylogenetic analysis of HB2018 and 207 PEDVs in Asia showed that most PEDV strains circulating in Asia after 2010 belong to genotype GII, particularly GII-a. The PEDV vaccine strain CV777 belonged to GI, and thus, unmatched genotypes between CV777 and GII-a variants might partially explain incomplete protection by the CV777-derived vaccine against PEDV variants in China. In addition, we found the S protein of variant strains contained numerous mutations compared to the S protein of CV777, and these mutations occurred in the N-terminal domain of the S protein. These mutations may influence the antigenicity, pathogenicity, and neutralization properties of the variant strains.

Corresponding author
Yongxiang Tian, tyxanbit@163.com

## INTRODUCTION

Porcine epidemic diarrhea (PED) is a high contagious and devastating disease resulting in the watery diarrhea in suckling pigs with high mortality and morbidity (*Zhang et al., 2019*). The causative agent of PED, the porcine epidemic diarrhea virus (PEDV), is an enveloped, single-stranded, positive-sense RNA virus belonging to the genus *Alphacoronavirus* in the family *Coronaviridae* (*Woo et al., 2012*). PEDV possesses a 28-kb genome which encodes seven proteins including ORF1a, ORF1b, spike (S) glycoprotein, ORF3 hypothetical protein, envelop (E) protein, membrane (M) protein and nucleocapsid protein (*Guo et al., 2019*). Among these proteins, the S protein plays a key role in interaction between the virus

and host cells. S protein consists of 1383-amino acids (*Aziz et al., 2008*), and amino acid changes in S protein may lead to antigenic variations and affect the virus virulence (*Gong et al., 2018*; *Suzuki et al., 2018*). Therefore, this protein is commonly used as an important target for analyzing genetic variations and molecular epidemiology of PEDV (*Hsueh et al., 2020*).

PED outbreaks have been reported continuously in China since 1973. PED was well controlled since administration of a CV777-derived vaccine (*Chen et al., 2019a*; *Wang, Fang & Xiao, 2016a*). However, recent outbreaks of PED in China since 2010 was due to the re-emergence of PEDV, and the continuous spread of the virus during the last 10 years has resulted in serious economic losses in the pig industry in Asian countries (*Yang et al., 2013*). In these outbreaks, inactivated vaccines and attenuated live vaccines, which were derived from CV777, were used to control the disease but neither of them provided effective protection (*Sun et al., 2012*; *Zhou et al., 2012*). Moreover, the virus has evolved since 2010 (*Guo et al., 2019*; *Hsu et al., 2018*; *Sun et al., 2019*), and acquisition of whole genome features of PEDV provides a convenient tool for the tracking of PEDV epidemiology (*Chen et al., 2019b*). In addition, virus isolation and genetic analysis allow investigations on the prevalence of epidemic strains and will provide information for diagnosis and vaccine developments (*Li et al., 2018*). In this study, we isolated a highly pathogenic PEDV strain HB2018 from a pig in a farm experiencing PED outbreaks in Hubei province, China, and determined its complete genome sequence. By comparing the HB2018 genome sequence with the sequences of 207 PEDV isolates circulating in Asia, which were publicly available in the Genbank data base, this study also aims to elucidate the evolutionary and genetic characteristics of PEDV currently circulation in different regions of Asia.

## MATERIALS AND METHODS

### Virus detection and isolation

In 2018, an outbreak of diarrhea occurred in a CV777-vaccinated pig farm (numbers of sows ≥ 100) in Hubei Province in China. Many pigs in the farm suffered from severe watery diarrhea, and some of them died. Samples of intestinal tissues were collected from dead pigs and sent to the Veterinary Diagnostic Laboratory of Hubei Academy of Agricultural Sciences in Wuhan, China, for diagnosis. Tissues were immersed with Dulbecco's modified Eagle medium (DMEM; Gibco, Grand Island, NY, USA), and were then homogenized using a QIAGEN TissueLyser II (QIAGEN, Dusseldorf, Nordrhein-Westfalen, Germany). The sample homogenates were then frozen at −80 °C and thawed for three times. After that, the supernatants were filtered through a 0.22-μm membrane and were harvested for RNA and virus isolation. Total RNAs were extracted using TRIzol (Thermo, Waltham, MA, USA) and were reverse transcribed to cDNA using a Thermo Scientific First Strand cDNA Synthesis kit (Thermo, Waltham, MA, USA). Viral nucleic acids were detected by RT-PCR assays using the cDNA as templates and the primers specific for PEDV (F: 5′-TTCGGTTCTATTCCCGTTGATG-3′, R: 5′-CCCATGAAGCACTTTCTCACTATC-3′), TGEV (transmissible gastroenteritis virus) (F: 5′-TTACAAACTCGCTATCGCATGG-3′,

R: 5′-TCTTGTCACATCACCTTTACCTGC-3′) and PoRV (porcine rotavirus) (F: 5′-CCCCGGTATTGAATATACCACAGT-3′, R: 5′-TTTCTGTTGGCCACCCTTTAGT-3′), respectively.

Vero cells (Purchased from ATCC, Manassas, VA, USA) were used for virus isolation. In brief, homogenate supernatants and trypsin (5 µg/ml) were inoculated into monolayers of Vero cells, which were then incubated in a 37 °C incubator supplemented with 5% $CO_2$. Cells with obvious cytopathic effects (CPEs) were harvested, thawed, and refrozen multiple times. The harvested virus suspension was then inoculated into newly prepared Vero cells for passages, and the propagation was continuously performed for 20 passages (F20). Virus RNA was extracted every five passage for RT-PCR detection of the virus nucleic acids.

## Virus titration and serum neutralization

Virus titers were measured on 96-well plates using 10-fold serial dilutions of culture supernatant in triplicate per dilution to determine the quantity of viruses required to produce CPEs in 50% of cells. After incubating for enough time, no more CPEs appeared, and TCID50 was calculated using the Reed-Muench method (*Reed & Muench, 1983*). The virus titer was also determined by plaque assay using Vero cells and expressed as plaque-forming units (PFU) per mL. The serum neutralization (SN) test was performed in 96-well plates with inactivated serum collected from the guinea pigs infected with the vaccine strain CV777. The virus was diluted in serum-free DMEM to make 200 TCID in a 50 µL volume, and mixed with 50 µL of 2-fold serial dilution serum. The mixture was added to cells cultured in 96-well plates and incubated at 37 °C for 1 h. After removing the mixture and thoroughly washing three times with PBS, the cells were incubated at 37 °C with 5% $CO_2$ for 2 days. Neutralization titers were calculated as the reciprocal of the highest dilution of serum that inhibits CPEs.

## Genome sequencing and annotation

Genomic RNA was extracted using the TAKARA RNA extraction kit (Takara, Kusatsu, Shiga, Japan) following the manufacture instruction. The quantity and quality of the extracted RNA were measured by using a Nanodrop spectrophotometer (Thermo, Waltham, MA, USA). The RNA was then subjected to reverse transcription for cDNA using a cDNA synthesis kit (Thermo, Waltham, MA, USA). Genome sequencing was performed with a paired-end library constructed by using a NEB-Next® DNA Library Prep Master Mix Set for Illumina (NEB, Ipswich, MA, USA) and subsequently sequenced on an Illumina NextSeq 500 with 2 × 150 paired end sequencing chemistry. After filtering, the clean reads were assembled using SPAdes v3.10.1 (*Bankevich et al., 2012*) and assembled sequences were mapped to the reference genome. The prediction of the genes and proteins were conducted with Prokka v1.12 and RAST Serve (http://rast.nmpdr.org) (*Aziz et al., 2008*). The complete genome sequence as well as its annotations were deposited into NCBI GenBank under the accession number MT166307.

## Comparative genomics and bioinformatical analysis

The NCBI data was search for "porcine epidemic diarrhea virus" and a total of 207 complete genome sequences were publicly available for PEDV isolates representing different parts

of Asia (See Table S1). All of these 207 sequences were downloaded for further analysis. The average nucleotide sequence identity between the genomes of HB2018 and CV777 was calculated by ANI calculator (*Goris et al., 2007*). Sequence alignments were performed using MAFFT v7.4.02 (*Katoh & Standley, 2013*). Nucleotide sequence similarity and the putative recombination sites was assessed by SimPlot v.3.5.1 (*Lole et al., 1999*), with a sliding window size of 500 bp, step size of 100 nucleotides, and 1,000 bootstrap replicates, using gap-stripped alignments and the F84 (ML) distance model. Phylogenetic trees based on complete genome sequences were generated by using MEGA X software with 1,000 bootstrapping (*Kumar et al., 2018*).The evolutionary history was inferred by using the Maximum Likelihood method and Tamura-Nei model (*Tamura & Nei, 1993*). Initial tree(s) for the heuristic search were obtained automatically by applying Neighbor-Join and BioNJ algorithms to a matrix of pairwise distances estimated using the Maximum Composite Likelihood (MCL) approach, and then selecting the topology with superior log likelihood value. The tree is drawn to scale, with branch lengths measured in the number of substitutions per site. A maximum likelihood tree was also generated using the BEAST 2 package (version 2.6.3) (*Bouckaert et al., 2019*). Gamma correction for site heterogeneity and the GTR model (*Gatto, Catanzaro & Milinkovitch, 2007*) were selected for the tree generation. Both of the trees were annotated and visualized by using the iTOL v.4 online tool (Interactive Tree of Life, http://itol.embl.de/) (*Letunic & Bork, 2019*). Single nucleotide polymorphisms (SNPs) between two genome sequences were determined by the MAUVE package (version 2.4.0) (*Darling et al., 2004*), and the coding effect of these SNPs were analyzed using a previously reported local Perl command (*Peng et al., 2016*). Protein structure was generated using SWISS-MODEL (https://swissmodel.expasy.org). Protein N-glycosylation sites were predicted using online software (http://www.cbs.dtu.dk/services/NetNGlyc/). Threshold values of greater than 0.5 and Jury agreement 9/9 were used for the high-specificity N-glycosylation sites determination (*Sagesser et al., 1997*).

# RESULTS

## Isolation of PEDV HB2018 and its genomic characteristics

RT-PCR detection of the viral nucleic acids revealed that the intestinal samples from pigs suffered and died from severe watery diarrhea were positive for PEDV but negative for TGEV and PoRV (Fig. S1). Through virus isolation and purification using Vero cells and determination of PEDV nucleic acids using RT-PCR, a PEDV strain was finally recovered and designated HB2018. The TCID50/0.1 mL value of HB2018 was $10^{5.3}$. The complete genome sequence of PEDV strain HB2018 was 28,138 bp in length. This 2.8-kb genome contained seven open reading frames (ORFs): ORF1a (nucleotide positions 281 to 12,634), ORF1b (positions 12,664 to 20,625), S gene (positions 24,782 to 25,456), ORF3 (positions 25,675 to 25,667), E gene (positions 25,437 to 25,667), M gene (positions 25,675 to 26,355), and N gene (positions 26,367 to 27,692).

Phylogenetic analysis based on the complete genome sequence showed that HB2018 was phylogenetically distinct from the vaccine strain CV777 (Fig. 1A). According to the genotyping system based on a full-length genomic sequence analysis (*Guo et al., 2019*;

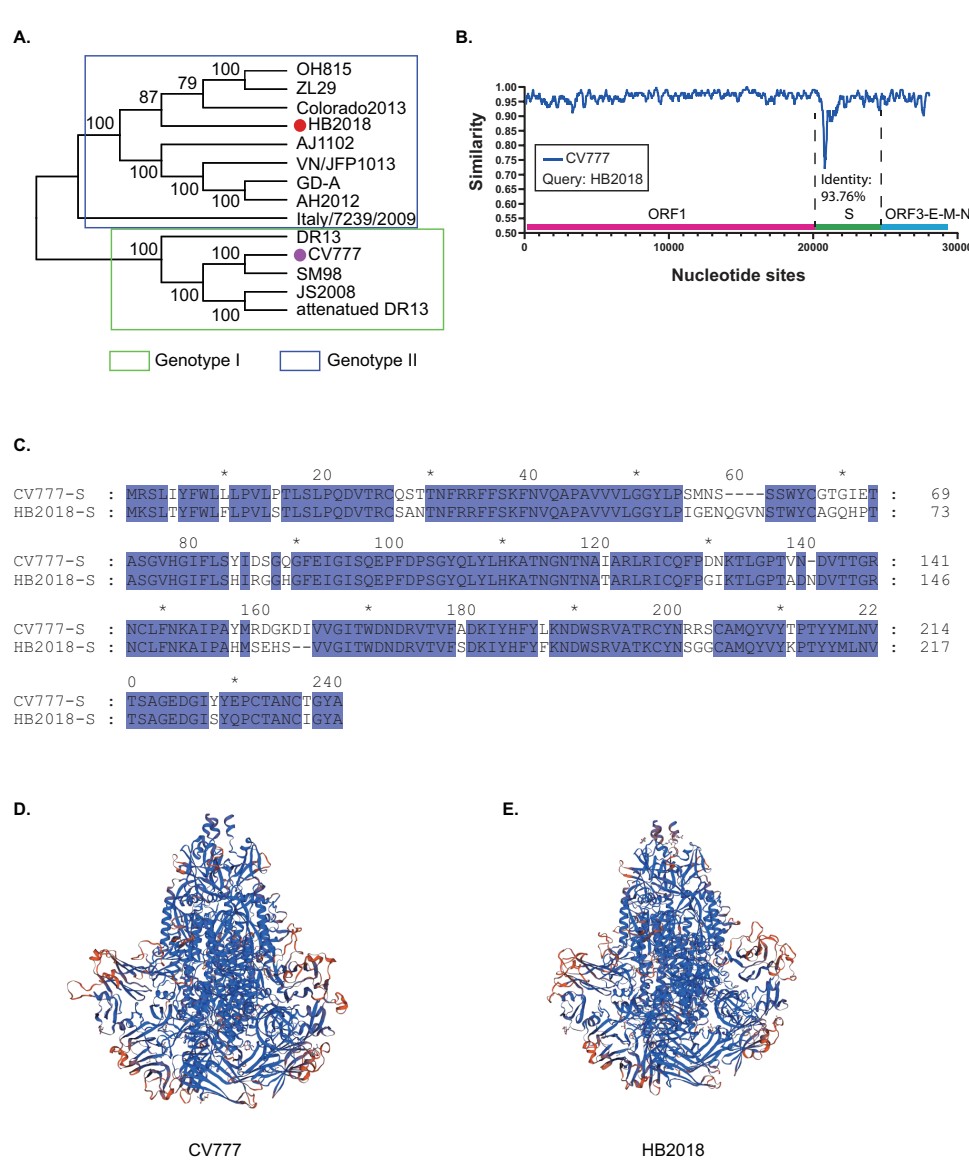

**Figure 1** **Phylogenetic and genetic characteristics of PEDV strain HB2018.** (A) Phylogenetic analysis of HB2018 and the other PEDV strains based on the whole genome sequence; (B) nucleotide similarity of the complete genome sequences between PEDV strains HB2018 and CV777; (C) sequence alignment of the S-NTD regions of PEDV strains HB2018 and CV777; (D) modelling the 3D structure of the S protein of CV777; (E) modelling the 3D structure of the S protein of HB2018.

*Wang, Fang & Xiao, 2016a*), HB2018 and CV777 belonged to two different genotype: HB2018 was assigned as a type GII strain while CV777 was a GI strain (Fig. 1A). The average nucleotide identity between the genomes of HB2018 and CV777 (GenBank accession no. AF353511) was 96.06% (Fig. S2). The ORF1, ORF3, E, M, and N genes of HB2018 as well as their encoding proteins were highly homologous to those of CV777 (nucleotide identity ≥ 95% for genes; amino acid similarity ≥ 95% for proteins) (Fig. 1B; Table 1). However, the identity of the S genes and proteins between the two strains was

**Table 1** Sequence comparisons of different ORF regions between HB2018 and CV777.

| ORFs | HB2018 vs. CV777 | |
| --- | --- | --- |
| | Amino acid similarity (%) | DNA identity (%) |
| ORF1 | 97.73 | 97.11 |
| S | 93.44 | 93.76 |
| ORF3 | 95.98 | 96.44 |
| E | 97.40 | 96.97 |
| M | 99.12 | 97.80 |
| N | 96.60 | 95.48 |

**Table 2** Single nucleotide polymorphism (SNP) analysis and dN/dS ratios of PEDV strains HB2018 and CV777.

| | ORFs | Sum | Non-synonymous | Synonymous | dN/dS |
| --- | --- | --- | --- | --- | --- |
| | Total | 925 | 262 | 663 | 0.395 |
| | ORF1a | 397 | 118 | 279 | 0.423 |
| | ORF1b | 188 | 26 | 162 | 0.160 |
| HB2018 vs. CV777 | S | 233 | 88 | 145 | 0.607 |
| | ORF3 | 24 | 9 | 15 | 0.600 |
| | E | 8 | 2 | 6 | 0.333 |
| | M | 15 | 2 | 13 | 0.154 |
| | N | 60 | 17 | 43 | 0.395 |

relatively low: the homology for nucleotide and amino acid sequences between HB2018 and CV777 were 93.76% and 93.44%, respectively (Figs. 1B & 1C; Table 1). SNP analysis determined a total of 946 SNPs in the genome sequence of HB2018 when compared to the genome sequence of the reference strain CV777. Among these SNPs, 925 SNPs including 262 non-synonymous substitutions and 663 synonymous substitutions were located with the ORF regions, with an overall ratio of nonsynonymous to synonymous substitutions (dN/dS) of 0.39 (Table 2). The dN/dS ratios in each of the ORFs encoded by the HB2018 genome ranged from 0.15 to 0.61, with the S protein had the highest dN/dS ratio (Table 2).

Compared to the S protein of CV777, the S protein of HB2018 had changes, deletions, and/or insertions of amino acids at multiple sites (Table S2 and Fig. S3). Notably, most of these mutations occurred in the N-terminal domain (NTD, 19-233aa) of the S protein (Fig. 1C; Fig. S3). Interestingly, some of these mutations were located within the neutralizing epitopes of PEDV (COE (499–638), SS2 (748–755), SS6 (764–771) and 2C10 (1368–1374)). In addition, these mutations led to a structural change at some parts of the HB2018 S protein compared to the CV777 S protein (Figs. 1D & 1E).

## Phylogenetic analysis of Aisan PEDV isolates

To explore the phylogenic relationships of the PEDVs currently circulating in Asia, we generated two maximum likelihood trees based on the whole genome sequences, either by using the MEGA X software with the Tamura-Nei model (Fig. 2A) or by using the BEAST 2 package with the GTR model (Fig. 2B). Both of the results revealed that the 208 PEDV strains in Asia, representing the 207 genome sequences publicly available in
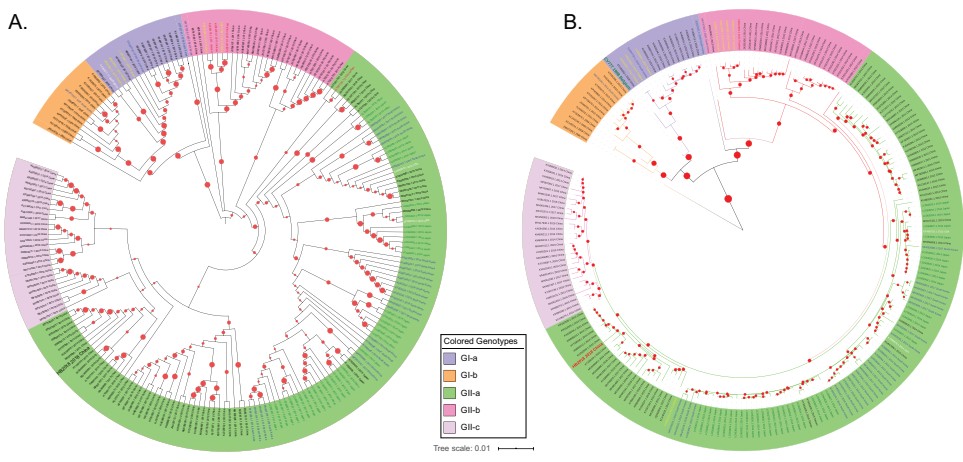

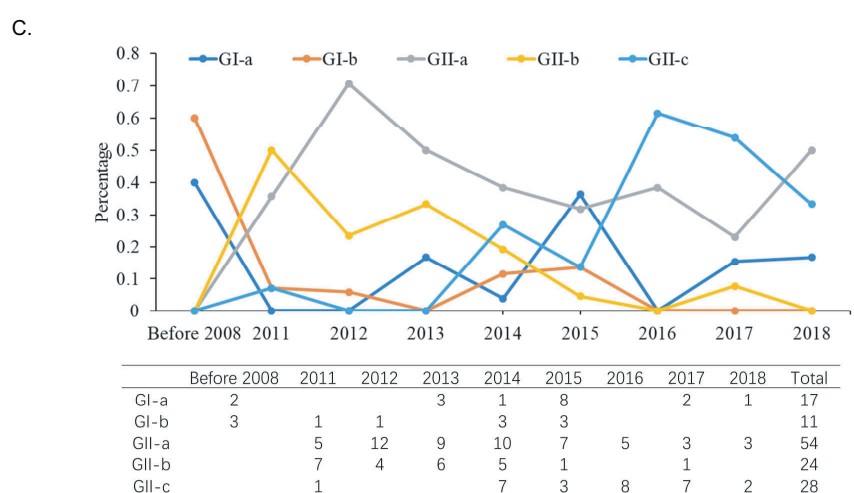

**Figure 2** **Genotyping of the 208 Asian PEDV strains and the non-Asian reference strains based on full-length genomic sequences.** (A) A phylogenetic tree was generated by using the MEGAX package. The evolutionary history was inferred by using the Maximum Likelihood method based on the TN93 model. The tree is drawn to scale, with branch lengths measured in the number of substitutions per site. GenBank accession numbers of strains, years, places of isolation, genogroups, and subgroups are shown. (B) A Maximum Likelihood tree was generated by using the BEAST 2 package. Gamma correction for site heterogeneity and the GTR model were selected for the tree generation. GenBank accession numbers of strains, years, places of isolation, genogroups, and subgroups are shown. (C) Line chart shows the number of PEDV sequences obtained by gene subgroup and year of sampling. Yearly percentages of samples positive for PEDV are indicated by different colored lines.

NCBI and the HB2018 sequence was divided into two genogroups: GI (classical) and GII (variant). Interestingly, isolates in China before 2010 and the vaccine strain CV777 were included within the GI genogroup. However, most of the PEDV isolates from China as well as the other Asian countries after 2010 belonged to GII genogroup (Figs. 2A & 2B). The phylogenetic trees also showed that the two genogroups consisted of several subgroups:
the genogroup GI was divided into two subgroups, GI-a and GI-b, while the genogroups GII was divided into three subgroups, GII-a, GII-b, and GII-c (Figs. 2A & 2B). The GI-a and GI-b subgroups included isolates from China before 2010 and several Chinese isolates between 2010 and 2015 (Figs. 2A & 2B). Most of the GII isolates from China and South Korea and all GII isolates from Japan were included within the GII-a subgroup, while less proportion of the Chinese GII isolates and most of the GII-a isolates from Southeast Asia (Vietnam and Thailand) were included within GII-b subgroup (Figs. 2A & 2B). Interestingly, the GII-c subgroup only consisted of isolates from China (Figs. 2A & 2B).

By analyzing isolation years and genogroups of PEDVs, the history of PEDV and the evolution in China are speculated. Between 1986 and 2008, only five PEDV strains were sequenced in China, and all of them belonged to G1 (Figs. 2A & 2B). However, the number of PEDV sequences increased significantly after 2010 (Fig. 2C). While several PEDV sequences belonged to genogroup GI after 2010, most sequences from China were GII strains (Figs. 2A–2C).

## Analysis on the S protein

Compared to the S proteins of the Chinese GI-a strains, amino acid changes, deletions, and/or insertions were observed at multiple sites within the S proteins of the Chinese GI-b strains (Table S3 and Txt S1). Compared to the S proteins of the Chinese GI strains, the S proteins of the Chinese GII strains commonly had amino acid changes, deletions, and/or insertions at several sites (Table S4 and Txt S1). Most these mutations occurred in S-NTD (19-233aa) of the S protein (Fig. 3A; Txt S1).

Compared to S proteins of the Chinese GII-a strains, S proteins of most isolates from Japan, South Korea, and Vietnam did not contain characteristic amino acid mutations, with the exception of S proteins of two Japanese strains (NIG-2/JPN/2014 and KMM-1/JPN/2014) which had a continuous deletion of 194 amino acids at sites 23–216 (Fig. 3B; Txt S2). In addition, four isolates from Japan (GenBank accession numbers LC063844–LC063847) and three isolates from South Korea (KNU-141112-S DEL5, KNU-141112-S DEL5ORF3, KNU-1406-1) had a continuous deletion of 5 amino acids (GENQG) at sites 56-60 in their S proteins compared to the S proteins of the Chinese GII-a strains (Fig. 3B; Txt S2). In addition to the GII-a strains, several isolates from China, South Korea, Thailand, and Vietnam were GII-b strains (Fig. 1). Compared to S proteins of most of the Chinese GII-b strains, S proteins of the GII-b isolates from Thailand and Vietnam had amino acid changes at sites 130-131 (SI →DN), 182 (Y →H), 287 (I →M), 324 (N →D), 327 (S →A), 358 (A →T), 367 (I →T), 433 (D →G), 558-559 (TN →PT), 1287 (E →K), and 1317 (L →F) (Txt S3).

The N-glycosylation sites in the S proteins of the Asian strains studied were investigated herein. Bioinformatical analysis revealed that most of the 208 Asian isolates contained 7∼9 high-specificity N-glycosylation sites in their S proteins. When combining all high-specificity N-glycosylation sites determined and deleting the duplicates, eleven sites appeared in most isolates, including 57NSTW60, 112NATA115, 127NKTL130, 212NVTS215, 320NDTS323, 347NSSD350, 510NITV513, 552NVTN555, 777NISI780, 1245NKTL1248, and 1257NRTG1260 (Table 3). Among these sites, 212NVTS215,

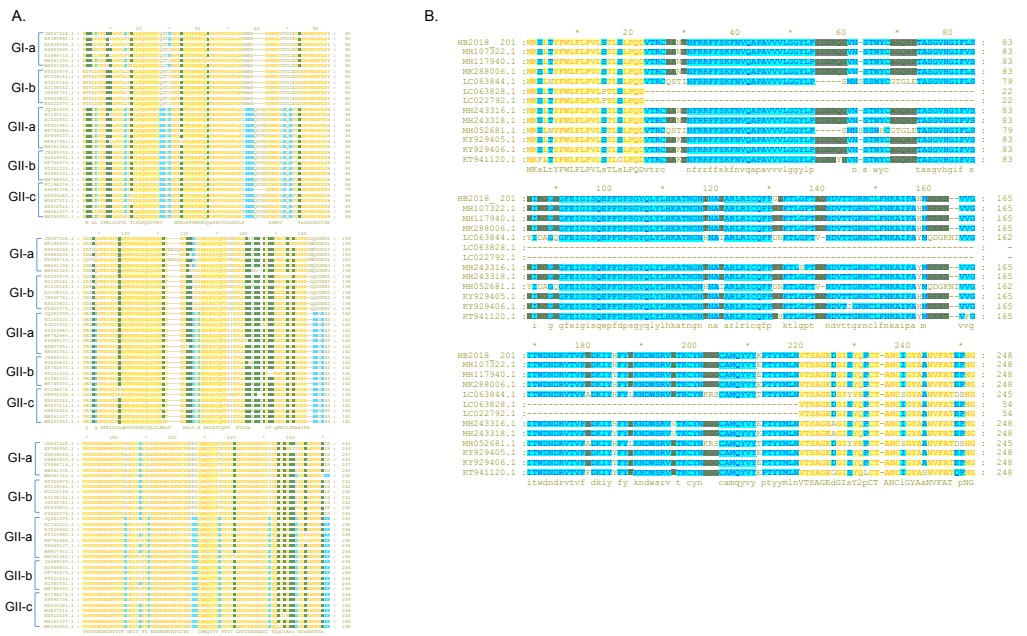

**Figure 3 Sequence alignment of the S-NTD regions of PEDV strains.** (A) Sequence alignment of the S-NTD regions of the strains from China; (B) sequence alignment of the S-NTD regions of the strains from Asia.

777NISI780, and 1245NKTL1248 were conserved in almost all of the 208 Asian strains. However, the N-glycosylation sites at sites 57–60 of some strains were "NSSS" rather than "NSTW". This is because the S proteins of these strains had the 59QGVN62 deletion compared to the S proteins of the other strains. Due to the amino acids changes, the N-glycosylation sites at sites 347–350 in some strains also changed from "NSSD" to "NSSN" or "NSTN". The 510NITV513 and 552NVTN555 N-glycosylation sites were missing in S proteins of the isolates from South Korea and Vietnam. Due to the large deletion of amino acids at sites 23–216, the 57NSTW60, 112NATA115, 127NKTL130, and 212NVTS215 N-glycosylation sites were missing in S proteins of two Japanese strains NIG-2/JPN/2014 and KMM-1/JPN/2014, and seven N-glycosylation sites, including 130NDTS133, 157NSSN160, 494NVTS497, 549NCTE552, 587NISI590, 1055NKTL1058, and 1067NRTG1070 were retained in these two strains.

## Analysis of the ORF3-E-M-N proteins

Unlike the S protein, ORF3 is a conserved protein among the PEDV isolates (*Wang et al., 2016b*). However, several PEDVs were found to have characteristic amino acid mutations in the ORF3 (Fig. 4). Compared to ORF3 proteins of many G1 strains and all GII strains, nine GI-a PEDVs (ZJUG12013, 85-7, 85-7-mutant 1, 85-7-mutant 2, 85-7-mutant 3, 85-7-mutant 4, 85-7-mutant 5, 85-7-A40, and 85-7-C40) had a continuous deletion of 70 amino acids at their N-terminal (positions 1–70) of ORF3; two GI-a strains (85-7-mutant 2 and 85-7-mutant 4) had a continuous deletion of 47 amino acids at their C-terminal (positions 178–224) of ORF3; while nine GI-b strains (JS2008, AH-M, SD-M, SQ2014,

Liang et al. (2020), *PeerJ*, DOI 10.7717/peerj.10114

**Table 3  High-specificity N-glycosylation sites predicted in Asian strains.**

| Country/ Regions | Strain | High-specificity N-glycosylation sites[a] | | | | | | | | | | | | | | | | | | | |
|---|---|---|---|---|---|---|---|---|---|---|---|---|---|---|---|---|---|---|---|---|---|
| | | 57 NSTW | 112 NATA | 127 NKTL | 212 NVTS | 320 NDTS | 347 NSSD | 510 NITV | 552 NVTN | 777 NISI | 1245 NKTL | 1257 NRTG | 22 | 347 | 382 | 421 | 524 | 739 | 869 | 1198 | 1274 |
| China | CV777 | – | NTSA | • | • | – | • | • | • | • | • | – | – | – | – | – | – | – | – | – | – |
| | HB2018 | • | • | – | • | • | • | – | – | • | • | – | – | – | – | – | – | – | – | – | – |
| | LZC | – | – | • | • | • | • | • | • | • | • | • | – | – | – | – | – | – | – | – | NLTG |
| | DR13 | • | • | – | • | • | NSSN | – | – | • | • | • | – | – | – | – | – | – | – | – | – |
| | SD-M | NSSS | NTSA | • | • | – | • | • | • | • | • | – | – | – | – | – | – | – | – | – | – |
| | AJ1102 | • | • | • | • | • | • | • | – | • | • | – | – | – | – | – | – | – | – | – | – |
| | FJZZ1 | • | • | • | • | • | • | • | – | • | • | – | – | – | – | – | – | – | – | – | – |
| | LS | – | – | – | • | • | • | – | – | • | • | • | NVTR | – | – | – | – | – | – | – | – |
| | CHS | – | – | • | • | • | • | • | • | • | • | • | – | – | – | NFTD | – | – | – | – | – |
| | CHHNQX-314 | – | • | • | • | • | • | – | – | • | • | • | – | – | – | – | NLTA | NCTE | – | – | – |
| | CHYJ130330 | • | • | – | • | • | NSTN | • | – | NITI | • | • | – | – | – | – | – | – | – | – | – |
| Thailand | CBR1 | • | • | • | • | – | • | • | • | • | • | • | – | – | – | – | – | – | – | – | – |
| | AVCT12 | – | – | • | • | • | • | • | • | • | • | – | – | – | – | – | – | – | – | NYTA | – |
| Taiwan | PT-P5 | • | • | – | • | • | NSSN | • | – | • | NKTR | • | – | – | – | – | – | – | – | – | – |
| South Korea | KNU-1709 | • | • | – | • | • | NSSN | – | – | • | • | • | – | – | NSTV | – | – | – | – | NISS | – |
| | KNU-1702 | NSSS | – | • | • | • | • | – | • | • | • | – | – | – | – | – | – | – | – | – | – |
| | KNU-1305 | • | • | – | • | • | NSSN | – | – | • | • | • | – | NFSL | – | – | – | – | – | – | – |
| Vietnam | HUA-14PED96 | • | • | – | • | • | NSSN | – | – | • | • | – | – | – | – | – | – | – | – | NISS | – |
| | VN/JFP1013 | • | • | • | • | – | • | – | – | • | • | • | – | – | – | – | – | – | – | – | – |
| Japan | Tottori2 | – | – | • | • | • | • | • | • | • | • | – | – | – | – | – | – | – | – | NYTA | – |
| | OKY-1 | – | – | • | • | • | • | • | • | • | • | • | – | – | – | – | – | – | – | – | – |
| | IBR-7 | • | • | – | • | • | NSSN | – | – | • | • | • | – | – | – | – | – | – | – | – | – |

**Notes.**

[a]The high-specificity N-glycosylation sites and their amino acids are summarized, the representative strains from each country/regions are listed. • means the strain has this high-specificity N-glycosylation site; –means the strain has no this high-specificity N-glycosylation site; the amino acid sequences means the strain has a high-specificity N-glycosylation in this sits , but the amino acid sequences are different with the common sequences.

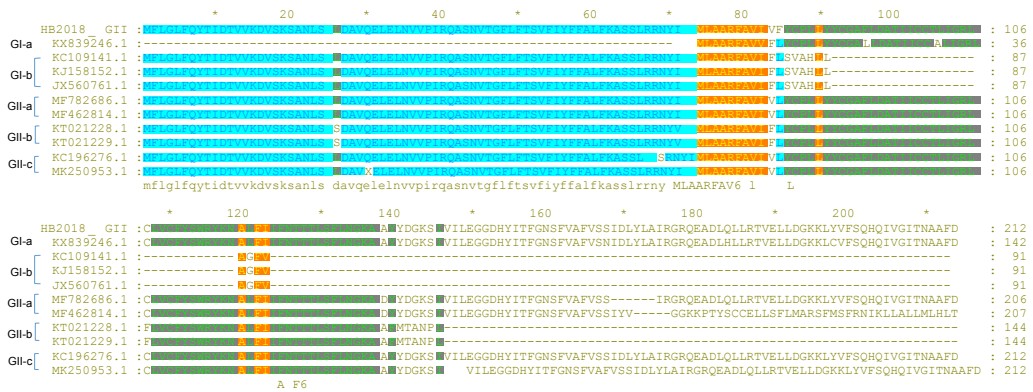

**Figure 4   Sequence alignment of ORF3 of PEDV strains.**

SC1402, HLJBY, PEDV-SX, JSLS-12015 and JS-22015) had a continuous deletion of 133 amino acids at their C-terminal (positions 92-224) of ORF3 (Fig. 4). Compared to ORF3 proteins of many GI and GII strains, ORF3 protein of a GII-a strain (CHSXYL2016) had a continuous deletion of 14 amino acids at positions 211-224; ORF3 protein of another GII-a strain (NW17) had a continuous deletion of 6 amino acids (DLYLAI) at positions 168-173; while ORF3 proteins of six GII-b strains (YN15, YN30, YN60, YN90, YN144, YN200) had a continuous deletion of 79 amino acids at their C-terminal (positions 146-224) (Fig. 4). Compared to ORF3 proteins of the GII-a isolates, more than half of the GII-b isolates had amino acid changes at positions 25 (L →S), 70 (I →V), 80 (V →F), 107 (C →F), 168 (D →N), and 182 (Q →H) (Fig. 4). Interestingly, these amino acid changes were also found in the ORF3 proteins of many GII-c isolates after 2016.

Sequence comparisons revealed that there were no common INDELs or mutations in E proteins of one subgroup of GII strains compared to E proteins of other subgroups of GII strains (Fig. 5A). M proteins of most GII-a strains had a glutamine (Q) at site 13; however, all GII-b strains isolated between 2011 and 2012 had a glutamic acid (E) at the same position in their M proteins, and this amino acid change (Q →E) occurred frequently in M proteins of GII-b since 2013 (Fig. 5B). A similar phenomenon was also observed in the M proteins of the GII-c strains, as most of the GII-c strains isolated before 2016 had a glutamine (Q) at position 13 in their M proteins, but a Q →E change at position 13 was seen in the M proteins of more frequently in strains isolated after 2016. In addition, amino acid changes at positions 192 (G →S) and 214 (S →A) appeared simultaneously in M proteins of some GII-b and GII-c strains. Similarly, in N proteins, amino acid changes at positions 216 (M →V) and 241 (R →K) appeared simultaneously in many GII strains (Fig. 5C).

# DISCUSSION

As an infectious virus attracted great intention, the PEDV strains were frequently reported and isolated in Asia. The virus isolation and genetic analysis will provide important information for PEDV research and vaccine developments. In this study, we isolated a

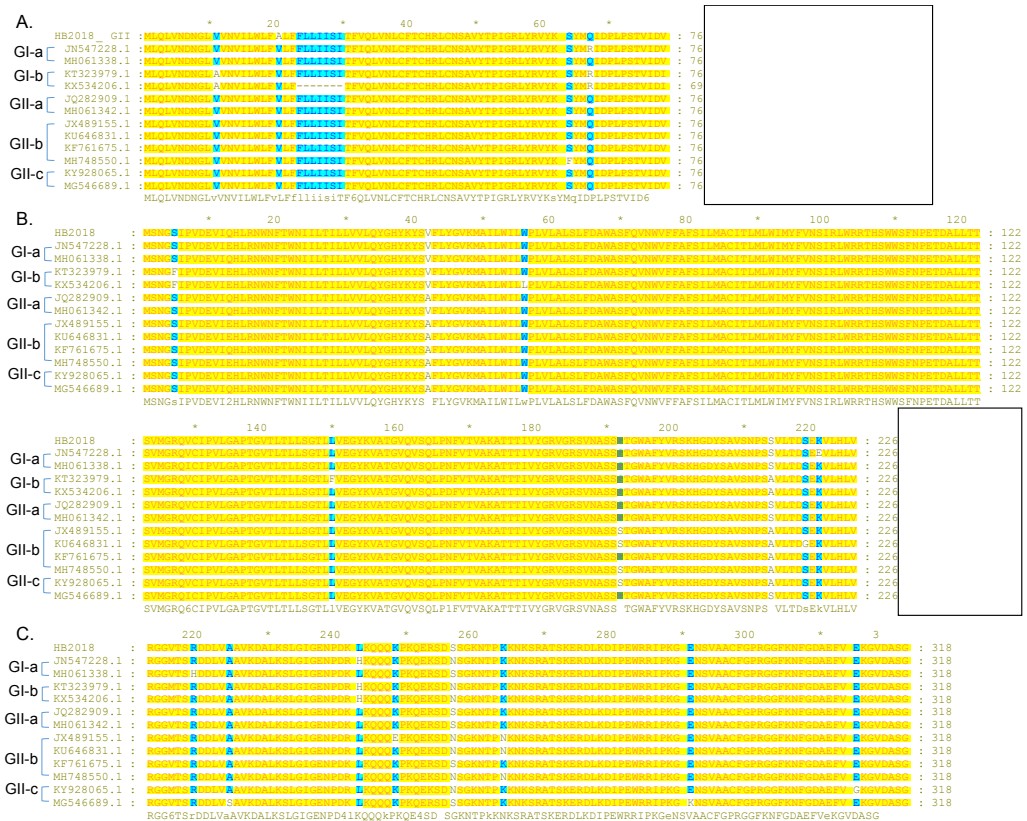

**Figure 5** **Sequence alignment of E, M and N protein of PEDV strains.** (A) E protein; (B) M protein; (C) N protein.

GII-a strain HB2018 and determined its genomic characteristics (Fig. 1A). Comparative genomic analysis revealed that the ORF1, ORF3, E, M, and N genes of HB2018 as well as their encoding proteins were highly homologous to those of CV777 (Fig. 1B; Table 1). However, a number of SNPs were determined within these ORFs, with the S proteins showed the highest dN/dS ratio (Table 2). Since dN/dS ratio is commonly used as a measure of purifying versus diversifying selection (*Rocha et al., 2006*), the highest dN/dS ratio suggests S protein is under diversifying selection, and this diversifying selection might be associated with its frequent interaction with host cells.

Sequence alignments determined many mutations in the genome sequence of HB2018 compared to that of the reference strain CV777. These mutations, especially in the S protein, might be the pathogenic determinants for it, because some deletions and insertions in the S protein may change the antigenicity, pathogenicity and neutralization properties (*Chen et al., 2019a*; *Sagesser et al., 1997*; *Zhang et al., 2015*). The presence of these mutations in the NTD of S protein in HB2018 might have an effect on the viral pathogenicity since the S-NTD domain is proposed to be the region relevant to the virulence of PEDV (*Hou et al., 2017*; *Su et al., 2018*; *Su, Hou & Wang, 2019*; *Suzuki et al., 2018*). In addition, the structural changes led by these mutations in S protein of HB2018 might influence the

immunogenicity. The distinct phylogenetic relationship between our isolation HB2018 and CV777 might partly explain why vaccination of pigs with CV777 did not provide effective protection against the infection of HB2018 in the vaccinated pig farm (Figs. 2A & 2B). The analysis based on isolation years and genogroups of PEDVs in Asia might also revealed the vaccine CV777 did not match with the pandemic PEDV isolations. Before 2010, all the strains in China belonged to GI (Figs. 2A & 2B). During this period, PEDV was well controlled in China due to the use of CV777 which was the GI-based vaccine (*Chen et al., 2019a*; *Yang et al., 2013*). The phylogenetic analysis of Asian PEDV isolates showed that most of the PEDV isolates from Asia after 2010 belonged to GII genogroup, while the vaccine CV777 were included within GI genogroup. These findings agree with the results of the other studies (*Guo et al., 2019*; *Wang, Fang & Xiao, 2016a*). The unmatched genotypes between CV777 and PEDV epidemic strains in Asia after 2010 could explain why vaccination with CV777 could not stop the outbreak of PED in many Asian countries after 2010 and provide effective protection against the current epidemic strains (*Chen et al., 2019a*; *Puranaveja et al., 2009*; *Sun et al., 2012*; *Zhou et al., 2012*).

With the most reported numbers of PEDV strains, China has more genogroups than other countries. The GII-c subgroup only consisted of isolates from China, these findings are also in agreement with previous studies (*Guo et al., 2019*; *Wang, Fang & Xiao, 2016a*), suggesting that the genotypes of PEDV strains circulating in China might be more heterogeneous than those of the isolates in other Asian countries. These findings may also explain why PEDV vaccines developed in China contain more than one strains that generally include CV777 and at least one more local GII isolate (http://vdts.ivdc.org.cn:8081/cx/#). The new emerged PEDV in 2010 might accelerate numerous isolations and sequencing of PEDVs (*Li et al., 2012*; *Yang et al., 2013*). In this article, it was found that the number of PEDV sequences increased significantly after 2010 and most sequences were GII strains. These results are in good agreement with the findings of the PEDV epidemiological investigations in China (*Chen et al., 2019a*; *Sun et al., 2018*). It has been reported that PEDV GII isolates were more virulent than GI isolates (*Vlasova et al., 2014*). This might in part explain why the traditional vaccines had no to little effect on the control and spread of PEDV in China after 2010. It is noteworthy that PEDV GII strains are also responsible for the recent outbreaks of PED in North America and Europe (*Choudhury et al., 2016*). These findings suggest the circulation of PEDV GII strains also pose a problem to the global pig industry.

S protein is the most variable protein of PEDV, the amino acid changes in this protein may lead to virus variation and affect the virus virulence (*Gong et al., 2018*; *Suzuki et al., 2018*). The mutations between were found between GI-a strains and GI-b strains, it is still uncertain whether these mutations between them has a biological significance. While, the mutations occurred in S-NTD of the S protein between GI strains and GII strains might in part explain why do the PEDV GII isolates be more pathogenic than the GI isolates (*Vlasova et al., 2014*), as S-NTD is proposed to be the region relevant to the virulence of PEDV (*Hou et al., 2017*; *Su et al., 2018*; *Su, Hou & Wang, 2019*; *Suzuki et al., 2018*). It is worthy note that PEDVs with insertions of amino acids at 167–168 and deletions of amino acids at 55–58 and 144 in their S proteins are called S-INDEL strains

(*Wang, Byrum & Zhang, 2014*). A previous study has found infection of the S-INDEL strains could induce pro-inflammatory cytokines through the non-canonical NF-κB signaling pathway by activating RIG-I; however, infection of the non-S-INDEL strains suppresses the induction of pro-inflammatory cytokines and type-I interferon production by down-regulation of TLRs and downstream signaling molecules (*Temeeyasen et al., 2018*). Whether the continuous deletion of 194 amino acids occurred in Japanese strains will affect the virulence of these strains are unknown and warrant further exploration. A previous study however has found that a Japanese strain Tottori2, which had the same deletion, had non-lethal effects in piglets (*Masuda et al., 2015*). The mutations also were found in C domain of S protein, since the NTD and C-domain both can bind to the host cell receptor and function as the receptor-binding domain, the amino acid changes in their sequences may have important role for the virus (*Li, 2012*). It has been reported the N-linked glycosylation sites on the S protein of some coronaviruses such as SARS-CoV play a critical role in the viral entry (*Han, Lohani & Cho, 2007*). The phylogenetic and N-linked glycosylation sites analysis of S protein may offer reasons for further studies. There was no too many mutations were found in the ORF3-E-M-N proteins, it might be because some of them, such as E protein, do not bear too much immune selective pressure since it has no effect on the host cell growth or cell cycle (*Xu et al., 2013*).

## CONCLUSIONS

In conclusion, through virus isolation and complete genome sequencing, we obtained PEDV HB2018 strain. Using this virus, we investigated the genetic and phylogenetic characteristics of PEDV isolates in China as well as in Asia in this study. Phylogenetic analysis revealed heterogeneous genotypes of PEDVs circulate in Asia, but GII particularly GII-a genotype represents the main epidemic genotype in the continent. Our study also revealed that most of the PEDVs currently prevalent in Asian countries displayed a different genotype as well as a distant relationship from the conventional vaccine strain CV777. This finding might explain why CV777-derived vaccine provided poor protection against PEDV epidemics (variant strains) since 2010. In addition, we also identified many mutations in the S, ORF3, E, M, N proteins of the variant strains (GII) compared to those of the classical strains (Temeeyasen et al). The presence of these mutations, particularly those determined in the S proteins, may affect the antigenicity, pathogenicity, and neutralization properties of the variant strains.

## ACKNOWLEDGEMENTS

The authors sincerely appreciate Dr. Zhong Peng at College of Veterinary Medicine, Huazhong Agricultural University, Wuhan, China for the language reversion and suggestions on the analysis.

### Funding

This work was supported by the Key Laboratory of Prevention and Control Agents for Animal Bacteriosis (Ministry of Agriculture) (grant number: KLPCAAB-YTP-1801), the open funds of the Key Laboratory of Preventive Veterinary Medicine of Hubei Province, and the China Postdoctoral Science Foundation (grant number: 2019M652609). There was no additional external funding received for this study. The funders had no role in study design, data collection and analysis, decision to publish, or preparation of the manuscript.

### Grant Disclosures

The following grant information was disclosed by the authors:
Key Laboratory of Prevention and Control Agents for Animal Bacteriosis (Ministry of Agriculture): KLPCAAB-YTP-1801.
Key Laboratory of Preventive Veterinary Medicine of Hubei Province.
China Postdoctoral Science Foundation: 2019M652609.

### Competing Interests

The authors declare there are no competing interests.

### Author Contributions

- Wan Liang conceived and designed the experiments, performed the experiments, analyzed the data, prepared figures and/or tables, authored or reviewed drafts of the paper, and approved the final draft.
- Danna Zhou conceived and designed the experiments, performed the experiments, analyzed the data, prepared figures and/or tables, and approved the final draft.
- Chao Geng conceived and designed the experiments, performed the experiments, prepared figures and/or tables, and approved the final draft.
- Keli Yang and Zhengying Duan conceived and designed the experiments, prepared figures and/or tables, and approved the final draft.
- Rui Guo, Wei Liu, Ting Gao, Ling Zhao, Dongwan Yoo and Yongxiang Tian conceived and designed the experiments, authored or reviewed drafts of the paper, and approved the final draft.
- Fangyan Yuan analyzed the data, authored or reviewed drafts of the paper, and approved the final draft.
- Zewen Liu performed the experiments, authored or reviewed drafts of the paper, and approved the final draft.

### DNA Deposition

The following information was supplied regarding the deposition of DNA sequences:
The whole genome sequence of PEDV HB2018 is available at GenBank: MT166307.

### Data Availability

The raw data are available in the Supplementary Files.

## Supplemental Information

Supplemental information for this article can be found online at http://dx.doi.org/10.7717/peerj.10114#supplemental-information.

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
