# Peer review of "Isolation and evolutionary analyses of porcine epidemic diarrhea virus in Asia"

_PeerJ, doi:10.7717/peerj.10114_

## Round 0.1 · original submission · Major Revisions

Our reviewers consider that your analysis should be improved and refined. I agree with their suggestions, especially regarding the rigorousness of the phylogenetic approach and the inclusion of dN/dS, positive selection sites, etc.

·

Basic reporting

The authors have adhered to the journal's basic reporting guidelines.

Experimental design

No comment

Validity of the findings

No comment

Additional comments

The paper describes the isolation and characterization of a virulent strain of PEDV from a pig farm in China. The paper is well structured and the methods are detailed and reproducible enough. The have reported a distinct strain of PEDV different from the vaccine strain with mutations in the S protein which may be responsible for incomplete protection from the vaccine strain. The paper is well written but there are some typographical errors that should be looked into and corrected.
.
SPECIFIC COMMENTS: Abstract line 25, the authors school recast the phrase as thus. In this study we the genetic characteristics of PEDV in Asia. The phylogenetic should be deleted, phylogeny is also a part of genetic analysis in this context.
Also one isolate reported from just one farm is insufficient for the strong conclusions drawn and should be included as a limitation in the study.
Figure legend 1 line 472 (B) The legend does not correspond with the figure, the legend states that Sequence alignment on the complete genomes of PEDV strains HB2018 and CV777. While the figure is a similarity plot of the two PEDV strains illustrated on SimPlot. The authors are advised to make the appropriate correction.

Reviewer 2 ·

Basic reporting

Dear editor

The paper by Liang et al brings a new PEDV genome recently isolated in China. I think it is an interesting study which uses phylogenetic analysis to call attention to the possible inefficiency of the current PEDV vaccine against viruses from other “genogroups”.

Overall, my major criticism is over the phylogenetic methods chosen by the authors. While I think that the distinction between strains HB2018 and CV777 (the “vaccine strain”) is clear enough to appear whichever method was used, there are several issues (see below) concerning the phylogenetic methods chosen by the authors.

These concerns are easy to tackle and only require more data analysis.

Major issues

- Phylogenetic analysis

1) I'm very concerned about the phylogenetic analysis performed here. NJ is a "quick and dirty" heuristic that usually recovers the minimum evolution tree. However, distance-based methods are usually worse than character-based methods (such as ML - or Bayesian methods, which also rely on a likelihood function). Based on the current dataset, this NJ tree is also hard to root. Two non-mutually exclusive alternatives would be a) include an outgroup, such as another related alphacoronaviridae, or b) use a molecular-clock method based on sample dates (such as Beast), which will generate a rooted tree. There is also no information about the genetic distance used and a justification for that (note that even in MEGA X there is a module to compare the fit of different evolutionary nucleotide models). Simple genetic distances such as F84, JC, K2P may be unable to account for multiple hits in the alignment, which may have profound impacts on tree topology. I do not question that HB2018 and CV777 do belong to different genotypes, neither that the low effectiveness of the PEDV vaccine has to do with this issue. However, I strongly suggest that that the authors reanalyze their data using a more rigorous phylogenetic framework.

2) Why does the tree in Figure 1 have so few terminals? How were the 207 sequences filtered out? Does DR13 belong to any "genogroup"?

3) What is the point of Figure 1C? Do the authors believe that S protein has a different underlying tree than the rest of the genome? If so, which process would have caused this? Recombination? Or is it just an artifact of a higher mutation rate which increases phylogenetic "noise" in the region? Or is it just noise due to the lower absolute number of mutations available for phylogeny estimation? How is it possible to discriminate among these alternatives?

4) Similarly, in Figure 2, there is no information about the ML methodology in the Material and Methods, no justification for the use of JC model, and no justification for the rooting position. Note that several genotypes are not clades. While this can be a real phenomenon, a poorly reconstructed phylogeny can also play a role in this.

5) Are "genogroup" assignments a result from this study, or was the phylogenetic structure in PEDV known? This could be given in Table S1. On the other hand, if it is an original result, how it was performed, given that genogroups are not clades in the phylogenetic tree? It is important to clarify this issue.

- Other

6) Please be aware of the use of "homology" when discussing "identity". Homology is a qualitative term that refers (in this case) to the origin of a given genome position. The position is homologous irrespective of the nucleotide present in two given sequences. Newly inserted positions may lack a homologous equivalent (though the ORF itself is homologous), but this is not what is being computed. Also avoid referring to "homology" in lines 152 and 154.

7) I wonder if modelling the 3D structure of the S protein and understanding its topology and polarity may give other insights about the effectiveness of a general vaccine against PEDV.

8) Overall, I think that the discussion part of the "results and discussion" section could be shortened, as some ideas are repeated several times (such as the inference that the CV777-based vaccine is ineffective against GII strains).

9) Most of sections "Analysis of the S protein" and "Analysis of the ORF3-E-M-N proteins" is only descriptive. There are some interesting insights, but they get lost amidst idiosyncratic substitutions of uncertain biological meaning. I suggest the authors to rephrase this section trying to resume and discuss only the most important results. The complete list of list of mutations can be given as a supplementary material.

Minor issues

line 52. substitute "China in 2010" for "China since 2010"
line 54. I'm no native English speaker, but the use of "nor" in this sentence sounds weird to me. I'd replace it by "and"
line 61. "...and determined its complete genome sequence"
line 63. I'd substitute this sentence, which is quite vague, by a more formal description of the study objectives
line 116. remove "of"
line 118. Are these complete genomes?
line 157. I see no reason to give the full list of S changes in the text
line 185. I find this sentence a little bit misleading. Please note that "ALL" isolates <2008 included in the analyses are only 5 isolates.
line 202. substitute "strains" by "strain"
line 204. Please rephrase this sentence
line 285. What is "relatively highly"?

Figures have low resolution and are hard to read (at least in the proof pdf)

Experimental design

no comment (see basic reporting)

Validity of the findings

no comment (see basic reporting)

Additional comments

no comment (see basic reporting)

Reviewer 3 ·

Basic reporting

The manuscript describes the results of PEDV HB2018 genome comparative analysis. The PEDV HB2018 has been isolated and sequenced, and the evolutionary relationships with PEDV strains have been investigated. The results are of interest but should be improved by additional analysis or data.

Experimental design

The PEDV HB2018 phylogenetic, statistical and evolutionary analysis has been done with widely used software and algorithms. However, taking into account a number of analyzed sequences some additional investigations should've been performed ( e.g. recombination analysis, pairwise matrix, minority variants, dN/dS, positive selection sites, etc). I believe that the genetic analysis of one viral sequence is no longer acceptable as an original article.

Validity of the findings

The number of figures should be minimized and the quality improved. For instance, in Figure 2, it is almost impossible to see the name of the PEDV strains on the circular tree. The authors also should refrain from using print screens as figures.
The authors argue that the mutations in PEDV HB2018 strain may affect the antigenicity, pathogenicity, and neutralization properties of the variant strains, however, no results confirming this proposition are presented. The results of the virus neutralization test or comparative growing curve will definitely benefit the study.

Additional comments

The authors present the genetic analysis of novel PEDV HB2018 strain, which is important for animal coronavirus research, but I would recommend to reconsider the manuscript as a short communication or a brief report.

---

## Round 0.2 · Minor Revisions

Please address the final request from reviewer #2 ("It is still not clear why did the authors use the TN93 model. I assume that they used the “model selection” module of MegaX, mas this should be stated explicitly, as well as the criteria for choosing this model (BIC, AICc, etc).") I would also appreciate it if you could add the dN/dS and positive sites requested earlier by reviewer #3, which I believe would strongly improve your paper.

·

Basic reporting

The authors have adhered to the journals reporting format and the paper is well structured. I have no comment.

Experimental design

No comment

Validity of the findings

No further comment

Additional comments

The article is well written the authors have satisfied all my concerns, no further comment

Reviewer 2 ·

Basic reporting

Dear editor

Thank you for the opportunity to evaluate the new version of Liang et al manuscript. Overall, I think that the authors did a great job in the revision and that this I a much improved version of the study.

However, I have two brief considerations to make:

Experimental design

1) It is still not clear why did the authors use the TN93 model. I assume that they used the “model selection” module of MegaX, mas this should be stated explicitly, as well as the criteria for choosing this model (BIC, AICc, etc).

Validity of the findings

2) I didn’t like the splitting between Results and Discussion in the new version. First, I think that there is a lot of discussion going on in the Results section (I think that whenever you put your results in context considering the literature you are “discussing” them). Second, I think that combining results and discussion improved readability, without the problems of repeating ideas in both sections.

---

## Round 0.3 · Major Revisions

Reviewer #2's requests for model selection are, in my view, extremely relevant. I was also disappointed that many of the additional analyses (" e.g. recombination analysis, pairwise matrix, minority variants, dN/dS, positive selection sites, etc)." ) requested by reviewer #3 in their original review remain unperformed, even though I called attention to that omission in my previous decisions.

Reviewer 2 ·

Basic reporting

Dear editor/authors

Even though I think that the main conclusions of the paper are robust to phylogenetic model choice, I don't buy the explanation that it is ok to use a given model because others have use it before. Model selection in Mega X takes less than 5 minutes to run, and would give the authors a better justification for their selected model. GTR is also a widely used model. Gamma correction for site heterogeneity is also widely used. Because a given model is adequate for other datasets, it doesn't mean it is adequate for THIS dataset, and that's why model selection should be used in phylogenetic studies.

There are other, much more complex issues concerning model selection, which I don't think are relevant here. However, I still believe that a minimal justification is needed. MEGA X provides a relatively easy way of doing this. However, I leave the final decision for the editor as I don't think this impacts the conclusions of the study.

Your sincerely

Experimental design

no comment

Validity of the findings

no comment

---

## Round 0.4 · accepted · Accept

Thank you for the additional analysis.